# Cryptic mitochondrial DNA mutations coincide with mid-late life and are pathophysiologically informative in single cells across tissues and species

Alistair P. Green [1,5], Florian Klimm [1,2,5], Aidan S. Marshall[1], Rein Leetmaa[1], Juvid Aryaman[1,2], Aurora Gómez-Durán [2,3], Patrick F. Chinnery [2] & Nick S. Jones [1,4] ✉

Ageing is associated with a range of chronic diseases and has diverse hallmarks. Mitochondrial dysfunction is implicated in ageing, and mouse-models with artificially enhanced mitochondrial DNA mutation rates show accelerated ageing. A scarcely studied aspect of ageing, because it is invisible in aggregate analyses, is the accumulation of somatic mitochondrial DNA mutations which are unique to single cells (cryptic mutations). We find evidence of cryptic mitochondrial DNA mutations from diverse single-cell datasets, from three species, and discover: cryptic mutations constitute the vast majority of mitochondrial DNA mutations in aged post-mitotic tissues, that they can avoid selection, that their accumulation is consonant with theory we develop, hitting high levels coinciding with species specific mid-late life, and that their presence covaries with a majority of the hallmarks of ageing including protein misfolding and endoplasmic reticulum stress. We identify mechanistic links to endoplasmic reticulum stress experimentally and further give an indication that aged brain cells with high levels of cryptic mutations show markers of neurodegeneration and that calorie restriction slows the accumulation of cryptic mutations.

Diseases of later life represent a formidable human challenge[1]. Existing hallmarks of ageing point to disparate but interconnected classes of causal factors leaving the search for unifying factors in humans open[2,3]. Somatic theories of ageing implicate DNA mutation as a possible cause —but nuclear mutation levels are possibly too low to be fully explanatory in post-mitotic tissue[2–4]. Mitochondria and mtDNA mutation have been implicated in ageing[5,6]. MtDNA mutations certainly increase in number with age; in tissue and cellular aggregates, and in single-cell

colony forming assays subject to selection, e.g.[7–10] with mutations predominantly created during mtDNA replication[11]. There is emerging evidence that pathological mtDNA mutations are harboured at high levels in immune cells[12], and that these mutations can be linked to a cell phenotype[13]. MtDNA mutator mice and cardiac and skin twinkle mutants[14–17] suggest that very high levels of mtDNA mutation can yield pro-geroid phenotypes (and this can be reversed[17]) but shorter-lived organisms and haploid mutator organisms yield a more nuanced

[1]Department of Mathematics & Centre for the Mathematics of Precision Healthcare, Imperial College London, South Kensington, London, UK. [2]Department of Clinical Neuroscience & Medical Research Council Mitochondrial Biology Unit, School of Clinical Medicine, University of Cambridge, Cambridge, UK. [3]MitoPhenomics Lab, Centro Singular de Investigación en Medicina Molecular y Enfermedades Crónicas (CiMUS), Universidade de Santiago de Compostela, Campus Vida Avenida Barcelona, A Coruña, Spain. [4]I-X Centre for AI in Science, Imperial White City Campus, London, UK. [5]These authors contributed equally: Alistair P. Green, Florian Klimm. ✉e-mail: nick.jones@imperial.ac.uk

picture[18–21]. There is a separate debate about the relative role of mutations that are inherited or developmental and de novo mutations[22] with evidence pointing to selective effects in inherited and developmental mutations[23]. Recently there has been an explosion of ageing-related single-cell transcriptomics data, though how the age of single cells is defined is unclear[24,25] with mixed evidence suggesting a link between ageing and gene-expression variance[25]. While single-cell transcriptomics has been shown to allow variant calling in nuclear and mtDNA[26,27] there have been no efforts to link single-cell transcription to cryptic mtDNA ageing.

In this work, we draw on single-cell data from over 140,000 cells, four mammalian species and seven tissues, using a mix of single-cell RNA and ATAC sequencing (scRNA-seq, scATAC-seq), we link ageing in post-mitotic tissues to an understudied type of mtDNA mutation (cryptic: those mtDNA mutations which are unique to single cells in a sample) which is invisible in aggregate. We give evidence that these mutations constitute the dominant fraction of tissue mtDNA mutations, accumulate in a manner which coincides with species lifespan and are consonant with new theory we develop. This theory predicts that mutations will reach functionally relevant heteroplasmies within human lifespan, and infers an mtDNA mutation rate consistent with existing literature values. Through new experiments and informatics, we find evidence that cryptic mtDNA mutations are linked to the expression of genes linked to disrupted proteostasis and immune/inflammatory response. Looking across rat, human, and mouse we find links between cryptic mutation and multiple hallmarks of ageing. We further find indications that in aged neurons the presence of cryptic mutations correlates with markers of neurodegeneration and that caloric restriction slows the accumulation of cryptic mutations.

## Results

### Cryptic mutation is predominant and its accumulation coincides with lifespan

It has been shown that scRNA-seq can be used for mtDNA mutation identification[26,28,29], and we further corroborate the validity of this technique (see Supplementary Discussion S6). We leverage mutational information gained from scRNA-seq and scATAC-seq to study the presence of mtDNA mutations at different ages.

The mtDNA heteroplasmy $h$ of a mutation is the proportion of mtDNA molecules in a cell that bear that mutation. Empirically, we assign a heteroplasmy value to each mutation based on the fraction of reads carrying that mutation (see Eq. (1) in the 'Methods' section). In the following, we exploit distributional and comparative approaches to ensure robustness to inevitable errors in sequencing and variant inference: in particular, we consider distributions of cryptic heteroplasmies where each mutant site is found in only one cell amongst all cells from a given donor and has an associated single cell heteroplasmy. A distribution of cryptic heteroplasmies can be found from a collection of cells by recording the heteroplasmies of all cryptic mtDNA mutations found in each cell in the sample and building a histogram showing the frequencies with which different heteroplasmies are observed. We term this distribution of cryptic heteroplasmies the 'cryptic' site frequency spectrum (cSFS) and it is a natural object from population genetics. By taking the distribution of these heteroplasmies, we can find the probability that a cryptic mutation, picked at random from the set of all cryptic mtDNA mutations in these cells, has a particular heteroplasmy (we use an extended notion of the site frequency spectrum and include homoplasmic (100% heteroplasmic) mutants (examples in Fig. 1c, f)).

We first examine five scRNA-seq datasets covering two species, three tissue types, and three sequencing techniques to demonstrate the wide relevance of our results[30–33]. Though these sequencing technologies provide differing levels of coverage of the mitochondrial genome (see Supplementary Fig. S20), we find that even when downsampling our high-coverage datasets to match the lowest coverage our

main conclusions are unchanged (see Supplementary Fig. S24). With single-cell-level data, we see how mutations which would be detectable at $h \geq 0.5\%$ heteroplasmy in a bulk sequencing experiment make up only ~9% of the mutations in a tissue (Fig. 1d). Of the ~91% of mutations that are at $h < 0.5\%$ in bulk heteroplasmy almost all (~94%) are only found in single cells.

We hypothesise, supported by theory (discussed later), that the cSFS will evolve with time gradually spreading to higher heteroplasmies (Fig. 1c). We compare the cSFS from human donors of different ages to see how it evolves through life and find that, consistent with the hypothesis, the further apart in age two individuals are, the larger the difference in their cSFS, as measured by the rank biserial correlation difference, RBC-difference, between pairs (see Fig. 1e, f each data point is a comparison between all the cells of two individuals, see Supplementary Fig. S25 for alternative metric comparisons; these results are preserved when restricting to a single cell type of the pancreas, see Supplementary Fig. S26a–c). The RBC difference of two cSFSs (A and B) is a measure of how likely a mutation sampled from A has a higher heteroplasmy than a mutation sampled from B (see 'Methods' section). To exclude possible errors from library preparation and sequencing, we only consider cryptic mutations with heteroplasmy $h > 10\%$ (see 'Methods' section and Supplementary Discussion S6).

Next, we identify cells in a single-cell human pancreas dataset[30] which carry mutations that are likely dysfunctional. For this, we compute each cell's mtDNA load of cryptic mutations $\mu^{10\%}$ as the sum of the heteroplasmies above 10% of all cryptic mutations which are not synonymous protein-coding mutants (see Eq. (2) in the 'Methods' section). We find that the mtDNA mutant load $\mu^{10\%}$ increases with age (see Fig. 1g). Furthermore, we observe that the standard deviation of the mtDNA mutant load increases with age, indicating a increasing age-associated cellular heterogeneity. The correlation $Corr(\mu^{h\%}, t)$ between age $t$ and mtDNA mutant load $\mu^{h\%}$ is significant for all heteroplasmy thresholds $h > 10\%$ (see Supplementary Figs. S28 and S29), indicating that the accumulation of cryptic mutations occurs already at a low-heteroplasmy threshold but is observable across a wide range of heteroplasmies.

To confirm that these results reflect the underlying mtDNA, we also examined a 10x single-nucleus ATAC-seq dataset from the aged human brain[34], offering an orthogonal sequencing modality, which supports our hypothesis that older individuals have a cSFS which is shifted to higher heteroplasmies (see Supplementary Fig. S19). We further examined a cross-species 10x scRNA-seq atlas of human, mouse, pig, and rat lung[35] and confirmed that young mouse, pig, and rat cells carry no high heteroplasmy cryptic mutations.

### Cryptic mutations reach physiologically relevant levels in a manner consonant with theory

Using an established forwards-in-time simulation approach from population genetics, the Moran model[36], we simulate how the cSFS should evolve with time if a cell starts with no cryptic mutations and gradually acquires them over a lifetime, Fig. 1c (see 'Methods' section). Our simulations predict that the cSFS histogram should have an amount of mutation at higher heteroplasmies that increases with age (Fig. 1c) and which has a characteristic age at which high-heteroplasmies are reached. In the Moran model mtDNA replicate and are eliminated in a manner that keeps the population constant; de novo mutations occur with a fixed rate and mutations spread through the population because of random birth and death. Using the Moran model, in the long-time limit, the expected time for the set of mtDNA in a cell to have a single common ancestor is proportional to the product of the number of mtDNA in the cell and the half-life of the mtDNA population: rapid birth–death or small populations lead to faster fixation of mutations. In order to develop a full theory, we fit the Moran model using an established backwards-in-time model from

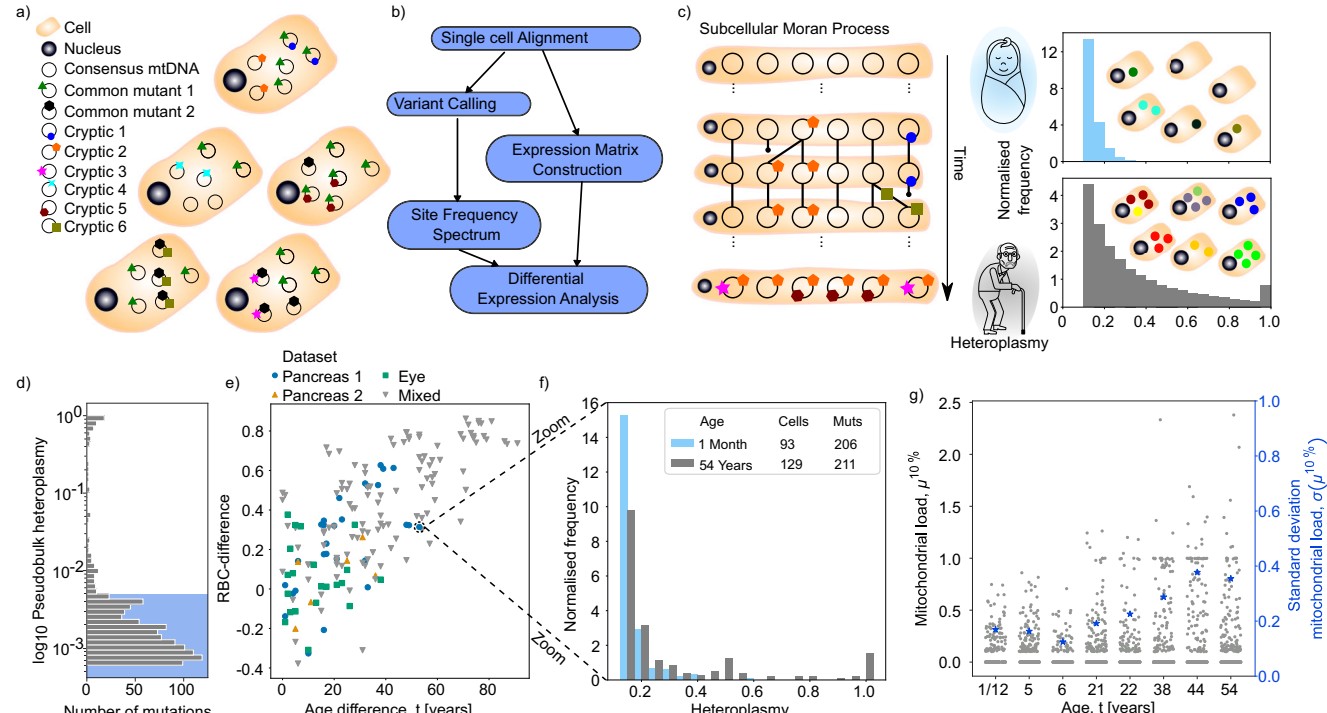

**Fig. 1 | Cryptic mtDNA mutation is a predominant form of mutation and reaches physiologically relevant levels in later life. a** Cells in tissues carry two mutant types−those common to multiple cells in the tissue and those unique to a single cell. **b** We exploit scRNA-seq and scATAC-seq on tissues taken from individuals of varying ages and pool variants from every cell allowing us to not only construct an expression matrix, but to examine the site frequency spectrum of a tissue and link changes in expression of single cells to inferred mtDNA mutant load. **c** Modelling sub-cellular population genetics with an out-of-equilibrium infinite-sites Moran model (left) we can predict how the normalised site frequency spectrum of cryptic mtDNA mutations (right) evolves over a lifetime and predict an accumulation of high heteroplasmy mutations in aged individuals. **d** Single-cell sequencing is necessary to reveal true mutant load of a tissue. In total, 90.8% of mtDNA mutations are at a pseudobulk heteroplasmy $h < 0.5\%$ (marked in blue) and so would not be reliably detected in most bulk experiments (data taken from all mutations found in ref. 30). **e, f** The cryptic site frequency spectrum (cSFS) evolves with time. We see that for 19 individuals across 3 human datasets taken from

different tissues and differing sequencing techniques, the rank biserial correlation distance (RBC-difference, a measure of how likely a mutation sampled from one spectrum will have a higher heteroplasmy than a mutation sampled from another) between two spectra increases with the age difference between the individuals (two-sided Spearman correlation $r \approx 0.70$ and $p < 10^{-26}$). By looking directly at an example pair of spectra (**f**) we see that, as predicted by our theory, there is an accumulation of mutants at high heteroplasmies (for a breakdown of the number of cells and mutations represented in the normalise cSFS see Table S5, for the cSFS of all donors see Supplementary Fig. S9). **g** The mitochondrial load $\mu^{10\%}$ of potentially pathogenic, cryptic mutations increases with age in human pancreas cells. We show the mitochondrial load $\mu^{10\%}$ for all eight donors in ref. 30 and observe an increase with age (two-sided Spearman correlation $r \approx 0.04$ and $p < 0.049$) and observe an even stronger effect for larger heteroplasmy thresholds (see Supplementary Discussion S8.5) The standard deviation $\sigma(\mu^{10\%})$ of the mitochondrial load also increases with age (blue asterisks; two-sided Spearman correlation $r \approx 0.88$ and $p < 0.005$).

populations genetics, the Kingman coalescent[37], which captures a broad class of forward models including the Moran model (see Supplementary Discussions S1 and S2 for details).

We used Bayesian inference to fit our model for the cSFS to the human datasets. In brief, our model fits a 'mitochondrial age', $W$, and scaled mutation rate, $\Theta$, to a set of cells from a donor tissue. Our model gives good fits to the cSFS of diverse individuals (see Supplementary Discussion S2.4 for full fitting details) and we find that the inferred mitochondrial age of each individual increases with the chronological age of each individual: providing a biological age marker for tissues and a candidate ageing clock[38] (Fig. 2a, these results are recapitulated using a single cell type in Supplementary Fig. S26). By making an assumption about the mtDNA copy number of cells, $N$, we can convert the inferred scaled mutation rate $\Theta$ to a mutation rate per base per replication, $\nu$ using the equation $\Theta = N\nu$. Under the assumption of 1000 mtDNA per cell, we infer a maximum a posteriori (MAP) estimate of the median mutation rate per base per replication of $4.6 \times 10^{-8}$, in accord with the literature[39] (Fig. 2b, see Supplementary Discussion S2 for details and full fitting results, and Supplementary Fig. S26 showing that the trends are recapitulated using only alpha cells from ref. 30).

A core model prediction is that the heteroplasmies of the cSFS increase until they reach a steady state, while the average number of

homoplasmies is constantly increasing. At steady state, heteroplasmic mutations are continually being lost or fixing at homoplasmy, while homoplasmic mutations accumulate, since they have no avenue to be lost from the population[40]. A key component of mitochondrial physiology, the 'threshold effect', is that when mtDNA mutations exceed a certain heteroplasmy (~60%) then buffering from wild-type mtDNA gene-products becomes harder and cellular effects of the mutation become strong; with marked effects associated with homoplasmic mutant mtDNA[41]. Experimental data in both human and mouse show a non-linear increase in homoplasmies with age, even across datasets and tissue types (Fig. 2d, this trend is also seen when restricted to a single cell type, see Supplementary Fig. S26b). Most strikingly, the data shows that after 80 years of age in humans, over 20% of cells carry a mutation with heteroplasmy $h > 95\%$ (Fig. 2d). We provide a line indicative of our theory, produced using the MAP estimate for ageing rate and mutation rate found using our model (see Supplementary Discussion S2 for details and full fitting results for both human and mouse data). Notably shorter-lived animals (mouse inset (Fig. 2d)) also reach high levels of homoplasmy much faster: reaching high numbers of homoplasmic cells coincides with these organism's lifespan. We also examine the first and second derivatives of the fit for number of homoplasmic mutations (Fig. 2e). These can be thought of as both the

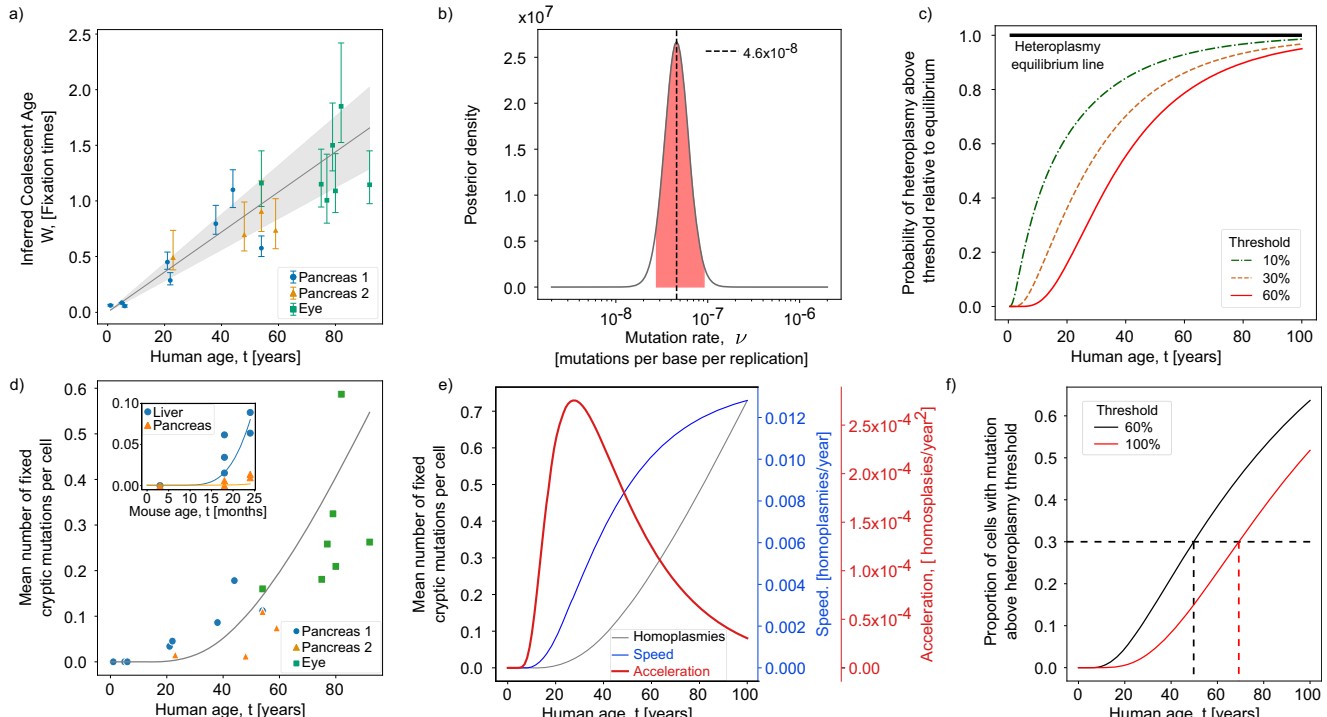

**Fig. 2 | Cryptic mtDNA mutations evolve in a clock-like manner consonant with theory and enable inference of mtDNA mutation rate. a** We present the 95% credible interval for each donor's mitochondrial age as a proportion of the expected time to fixation, and show the maximum a posteriori (MAP) estimate for regression, along with the 95% credible interval of the median inferred coalescent age (see Supplementary Discussion S2). **b** The posterior on the inferred mutation rate per base per replication is shown under the assumption of 1000 mtDNA per cell, with the MAP estimate highlighted as $4.6 \times 10^{-8}$. **c** The proportion of heteroplasmic mutations above a certain threshold reaches equilibrium at the same timescale as human life when modelled using the MAP parameter estimates. **d** The number of homoplasmic mutations (normalised by bases observed) per cell increases with age in humans (Spearman correlation $r \approx 0.89$ and $p < 10^{-7}$). Also shown is a line generated with the MAP parameters found from fitting the data to

our model (Supplementary Discussion S2). We see a similar accumulation of homoplasmic mutations in mice in samples from 2 tissues and 16 mice (Spearman correlation $r \approx 0.85$ and $p < 10^{-4}$). **e** We look at the first and second derivatives of the MAP estimate for number of homoplasmic mutations against time. These can be roughly equated to the speed and acceleration of ageing. The peak in acceleration of ageing occurs at around 40 years old. **f** Using the MAP estimate we look at the proportion of cells carrying a mutation above a certain threshold. After around 20 years cells begin to have mutations above a 60% heteroplasmy threshold and by age 40 an appreciable fraction of cells are carrying a mutation with heteroplasmy above 60% and cells have begun to accumulate mutations at homoplasmy (100% heteroplasmy). By the age of 80 nearly 30% of cells are predicted to have a homoplasmic mutations and nearly half are predicted to have mutations at a heteroplasmy above 60%.

speed and acceleration of mtDNA ageing, respectively. The shape of the speed of cryptic mtDNA ageing is roughly sigmoidal, with the ageing speed only beginning to increase after around 20 years. The acceleration of ageing peaks at around 40 years old when many of the more obvious signs of ageing have begun to present themselves. As noted the heteroplasmies in the cell (unlike the homoplasmies) do reach equilibrium and this timescale (using the same parameters as in Fig. 2d) coincides with lifespan (Fig. 2c). We can also look at the fraction of cells carrying a mutation above a certain heteroplasmy threshold. Again using the MAP estimate of model parameters we see that by ~50 years old, over 30% of cells are expected to carry at least one mutation at a heteroplasmy >60%. The time at which 30% of cells are expected to carry at least one mutation at homoplasmy is ~70 years old (Fig. 2f). These predictions are only slightly higher than our observed values: accounting for coverage differences for donors between 50 and 60 years old, 22 % of cells are expected to hold a mutation at a heteroplasmy >60%, and between 70 and 80 years old 23% of cells carry a homoplasmic mutation. These two thresholds demonstrate the multiple timescales at which mtDNA mutations can accumulate in tissues. We note that a third timescale is implied by this evidence: by ~20 years old, mutations at frequencies <10% have mostly reached their life-long levels in humans. While it is reasonable to expect a linear accumulation of nuclear DNA mutations with time the accumulation of cryptic heteroplasmies is non-linear and has

timescales that, remarkably, coincide with human ageing (see Supplementary Discussion S8.4 for further details).

## Cryptic mutations, unlike other types, can expand neutrally

Access to the cSFS of single cells allows us to examine selection against pathogenic mutations in more depth than the non-synonymous/synonymous mutant ratio. We can assign each protein-coding mutation a pathology class of either synonymous, low pathogenicity, or high pathogenicity (see 'Methods' section) and then examine the cSFS of the three pathology classes. By comparing this with a more conventional measure of selection, the non-synonymous/synonymous ratio, we can look in more detail at whether selection effects are dependent on the heteroplasmy of mutations. We perform this analysis on cryptic mutations taken from a high-quality full-length scRNA-seq data of healthy pancreas cells[30] and find that, for mutations above 10% heteroplasmy, there is no evidence of a significant shift in the cSFS of low or high pathogenicity mutations when compared to synonymous mutants (Fig. 3a), and the non-synonymous/synonymous ratio is not significantly shifted from 1: mutations that reach a heteroplasmy of >10% do not show evidence of selection. We repeat this analysis considering all non-cryptic mutations and find that as well as having a non-synonymous/synonymous ratio significantly shifted below 1, the SFS of the average heteroplasmy of non-cryptic, highly pathogenic mutations are also significantly shifted to lower heteroplasmies, which

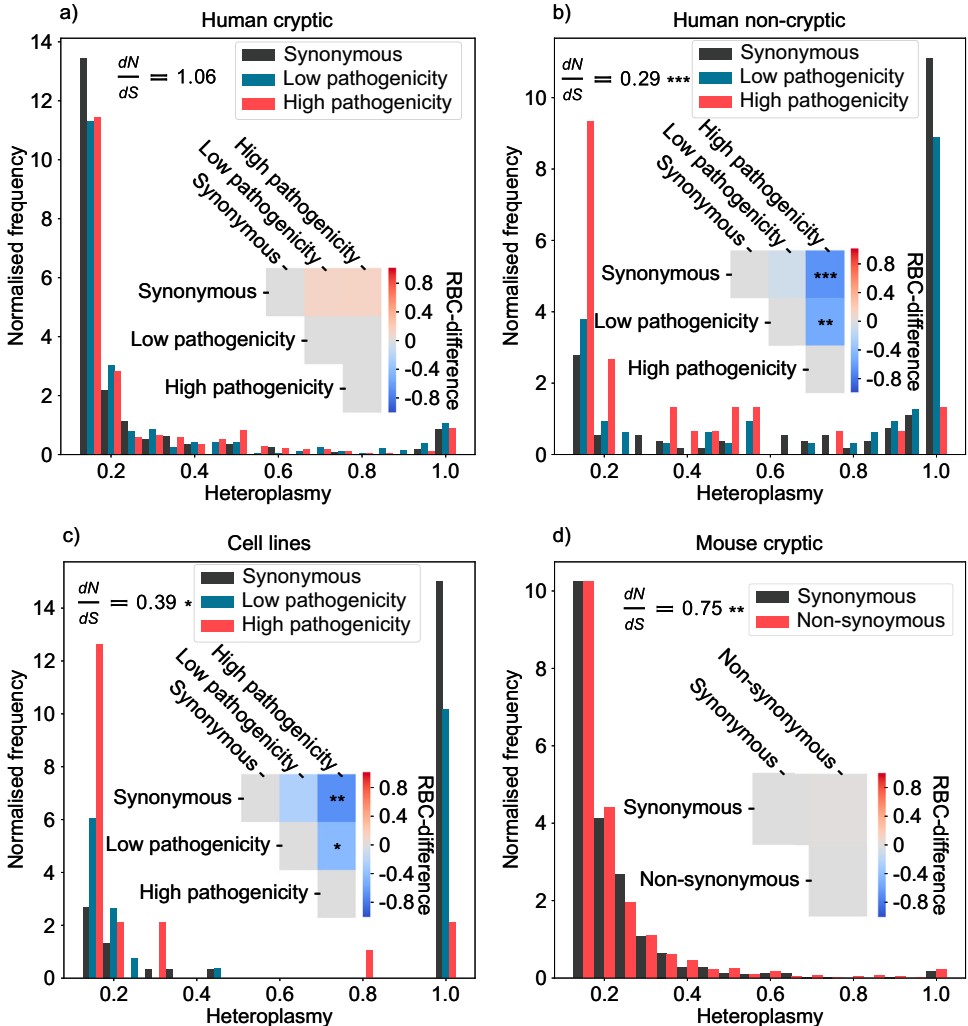

**Fig. 3 | Cryptic mtDNA mutations, unlike other types, can expand neutrally.** We show the SFS for mutations classified by their potential pathology, as well as the RBC-difference (see 'Methods' section) which shows the magnitude of difference between these SFSs (for a breakdown of the number of cells and mutations represented in each SFS, see Table S5). **a** All cryptic mutations taken from healthy human pancreas tissue[30] show no significant evidence of selection in either the non-synonymous/synonymous ratio or in the RBC-difference between cSFSs, indicating these mutations are not under selective pressure. **b** When we consider all non-cryptic mutations, we see a non-synonymous/synonymous ratio significantly < 1 (two-sided Fisher's exact test $p < 10^{-8}$) as well as a significant shift of the SFS of high pathogenicity mutations as compared to synonymous and low pathogenicity mutations (two-sided Mann–Whitney $U$ test $p < 10^{-6}$, $10^{-3}$, respectively), though we do not see evidence for selection against low pathogenicity mutations compared to synonymous, suggesting that mutations occurring on the germline or during development experience selective pressure based on the level of dysfunction they cause. **c** The SFS for all mutations found in cells from culture[43] show a similar selection pattern to non-cryptic mutations taken from tissue samples. They have a non-synonymous/synonymous ratio significantly < 1 (two-sided Fisher's exact test $p < 10^{-3}$) and the spectra of high pathogenicity mutations show a significant shift towards lower heteroplasmies when compared to synonymous and low pathogenicity mutations (two-sided Mann–Whitney $U$ test $p < 10^{-4}$, 0.05, respectively). **d** Though we cannot score the potential pathogenicity of non-synonymous mutations in mice, we look at the difference in the cSFS of synonymous and non-synonymous mutations finding no evidence of selection (two-sided Mann–Whitney $U$ test $p > 0.05$). We do, however, find a significant lack of non-synonymous mutations at heteroplasmies $h > 10\%$ (two-sided Fisher's exact test $p < 10^{-4}$).

is consistent with evidence of selection happening along the human germline and in development in a way which is modulated by the degree of pathogenicity the mutation causes[42] (Fig. 3b). Due to the lower coverage of the other human datasets[31,32] we did not have enough mutations to sufficiently power the fisher test for dN/dS ratio or the Mann–Whitney $U$ test for differences in the cSFS.

For comparative purposes we perform this analysis on scATAC-seq data for ENCODE cell lines[43] (GM12878 lymphoblastoid cells, K562 lymphoblast cells, and H1 human embryonic stem cells) and find that, analogously, the SFS of all mutations with both high and low pathogenicity are significantly shifted to lower heteroplasmies compared to synonymous ones, indicating selection against high heteroplasmy pathogenic mutants (Fig. 3c).

Though we cannot score the potential pathogenicity of mtDNA mutations in mice, we can compare the cSFS of synonymous and non-synonymous mutations above 10% from both liver and pancreas (Fig. 3d): we find no evidence of a shift in the cSFS between these two classes (as we did in Fig. 3a). We do, however, discover a significant shift in the non-synonymous/synonymous ratio below 1, and find evidence for analogous behaviour in human brain tissue (see Supplementary Fig. S19). This is consistent with a model where, for some tissues, some non-synonymous mutations undergo strong negative selection at heteroplasmies <10% (perhaps selective mitophagy) but where mutations that evade this selective mechanism expand neutrally in a manner independent of their heteroplasmy. The notion of selective mitophagy failing in somatic tissues has been observed in multiple

species[6,44], and while it is known that there are mechanisms by which deletions can expand in muscle fibres[45,46], these studies have been limited in their ability to detect a similar expansion of point mutations due to their PCR fragment size-based approach. Taken together this suggests there is more to be done to understand the role of selective mitophagy in somatic tissues.

These results, taken with the life-time evolution of the cSFS (Fig. 1d, f, h), point to the potential for accumulation of cryptic pathogenic mtDNA mutations through life, causing mosaic dysfunction in post-mitotic tissues.

## Cryptic mutation links to cellular phenotype in a manner consonant with markers of ageing pathophysiology

The number of cells with evidence of high-heteroplasmy cryptic mutations increases nonlinearly with age. To identify which genes' expression levels might be perturbed by the presence of cryptic mutations, we perform a differentially expressed genes (DEG) analysis: comparing cells with detected cryptic mutations which are not synonymous above 10% heteroplasmy and those without (see 'Methods' section) for the full-length scRNA-seq pancreas data[30]. After multiple-testing correction, we find 1342 genes significantly differentially expressed (see Fig. 4a), consonant with a possible large-scale transcriptional perturbation induced by cryptic mutations. First, as expected, we find mitochondrially encoded (e.g., MT-CO2, MT-RNR1) and nuclearly encoded (e.g, NDUFA1, NDUFA13) OXPHOS genes upregulated, which is an established response to impaired energy production[47]. Second, we identify genes associated with innate immune signalling and altered proteostasis (HSPA5, HSPA13, HSP90B1, and YME1L1[48]) downregulated and key inflammatory cytokine MIF upregulated[49].

Surprisingly, we identify an altered expression of long noncoding RNAs (lncRNAs), such as AC145207.3 and MIF-AS1, which have been hypothesised to play a role in ageing[50] and cancer cell proliferation[51]. While we here aggregate data from all donors to extract even subtle changes in gene expression, we also observe a similar perturbation of gene expression when performing this analysis at the level of single donors or cell types (see Supplementary Discussions S3 and S4).

Since proteins fulfil their biological functions through interaction, we then use SCPPIN[52] to integrate the $p$ values of differential expression with protein–protein interaction data and obtain a mutation-linked functional module consisting of 33 proteins (see Fig. 4b). This module is associated with endoplasmic reticulum (ER) stress in response to unfolded proteins and is consistent with mtDNA mutations yielding misfolded proteins, which trigger an ER-stress response (as is known for ageing-associated diseases of various tissues[53,54], including Alzheimer's disease). We find multiple Transmembrane emp24 domain-containing proteins to be perturbed, which hints at a dysregulated immune response[55]. TRIM25, a ubiquitin ligase that regulates the innate immune response, is at the centre of the module, highlighting the interplay between mtDNA mutation, immune response, and ER-stress[56–58].

We repeat this DEG analysis in seven scRNA-seq datasets from three different mammals (human, mice, and rat) and four different tissues, and identify in each of them that the cryptic mutations are linked to gene expression changes (see Fig. 4c, 'Methods' section, and Supplementary Figs. S32–38). We identify biological pathways linked to the presence of cryptic mutations across organisms by performing a GO-term enrichment analysis with PANTHER[59], highlighting terms that are enriched across at least six of seven datasets, confirming the broad applicability of our results across species and tissues (see Fig. 4d). Cryptic mutations coincide with a perturbation to the regulation of biological quality, response to stress, ER-stress, viral response, leucocyte activation, apoptosis, hypoxia and proteolysis, and protein folding. Combined with an enrichment of immune effector process these

terms are consistent with an immune response triggered by cryptic mutations: in line with recent findings linking neo-epitopes to de novo mtDNA mutations[56,58,60]. An enrichment of response to nutrient levels indicates that these processes might interplay with dietary interventions, a hypothesis that we test in the following section. Beyond genomic instability (a hallmark of ageing) the transcriptional discrepancies between cells with cryptic mtDNA load and those without are consonant with four further hallmarks of ageing (loss of proteostasis, deregulated nutrient-sensing, mitochondrial dysfunction, and altered intercellular communication).

Since the cryptic mutations we observe are diverse (and unique to each cell in the sample) we expect them to each create distinctive modulations to gene expression: nonetheless, Fig. 4d suggests common patterns, which can be accounted for as follows. In mitochondrial physiology, it is uncontroversial that mitochondrial-disease-related mtDNA mutations (e.g., LHON, MELAS) can cause changes in gene expression (we find LHON-associated mutations in 39 of the cells in the long-read human pancreas data[30]). The changes in Fig. 4d are consistent with the perturbations already known to be created by mitochondrial-disease mutations: changes related to hypoxia, ETC, ER-stress, and protein folding[61]. It is known that Complex I mutations have a marked effect in mitochondrial disease[62]—we find that mutations in the mt-ND4 and mt-ND5 genes are strongly associated with changes in cellular gene expression. The former being linked to 'response to unfolded protein' and the latter linked to 'ATP synthesis coupled electron transport' (see Supplementary Fig. S40). This points to a picture of ageing as partly a mosaic of different single-cell mitochondrial diseases.

Following our observations regarding complex I mutations we developed human cybrid cell lines allowing us to study the effect of different mtDNA mutations on the same osteosarcoma 143B $\rho^0$ nuclear genetic background. These cybrids contained mtDNA mutations at homoplasmy (Fig. 5a, with background matched controls) that we had separately identified as cryptic complex I mutations in our analysis of single-cell data: these lines thus allow us to functionally explore the effect of exemplar cryptic mutations (while, crucially, carefully controlling for other mtDNA mutations). We observe mutation-specific ETC (Fig. 5b) and mitochondrial ATP deficiency (Fig. 5c) with elevated ROS levels (Fig. 5d) and bioenergetic differences (Fig. 5e) in m.11778G > A. Transcriptomic analysis shows a pronounced effect on gene-expression in both cybrid lines (Fig. 5f, g). Pathway enrichment analysis yields evidence for ER-stress and altered proteostasis (Fig. 5g, e.g. AFT6, XBP1(S) and IRE1alpha) consonant with our previous observation (Fig. 4d) and we corroborated this by finding experimental evidence for the activation of Eukaryotic translation initation factor 2 alpha, a key kinase that responds to ER-stress[63] (Fig. 5h).

## Implications for calorie restriction and disease

Given the possible causal link between mtDNA mutation and ageing we asked first whether levels of cryptic mtDNA mutations could be controlled through an established anti-ageing technique, caloric restriction, and second, whether we could identify links between cryptic mutation and neurodegeneration (via single-nucleus-RNA-seq).

**Caloric restriction.** Caloric restriction has been recognised as one of the most effective interventions to promote longevity, and combat ageing[64]. We use scRNA-seq data from young and old ad libitum fed rats (Y-AL and O-AL, respectively) and old calorically restricted rats (O-CR)[65] to obtain cSFS of each group's liver and brown adipose tissue. Using the RBC-difference (see 'Methods' section) we observed that (as anticipated in Fig. 1f) cryptic mutations are of higher heteroplasmy in the O-AL than Y-AL (Fig. 6a, b), in both liver ($p < 0.0001$, Bonferroni corrected) and brown adipose tissue ($p < 0.001$, Bonferroni corrected). Likewise, mutations in O-AL are higher in heteroplasmy than in O-CR, in both liver ($p < 0.01$, Bonferroni corrected) and brown adipose tissue

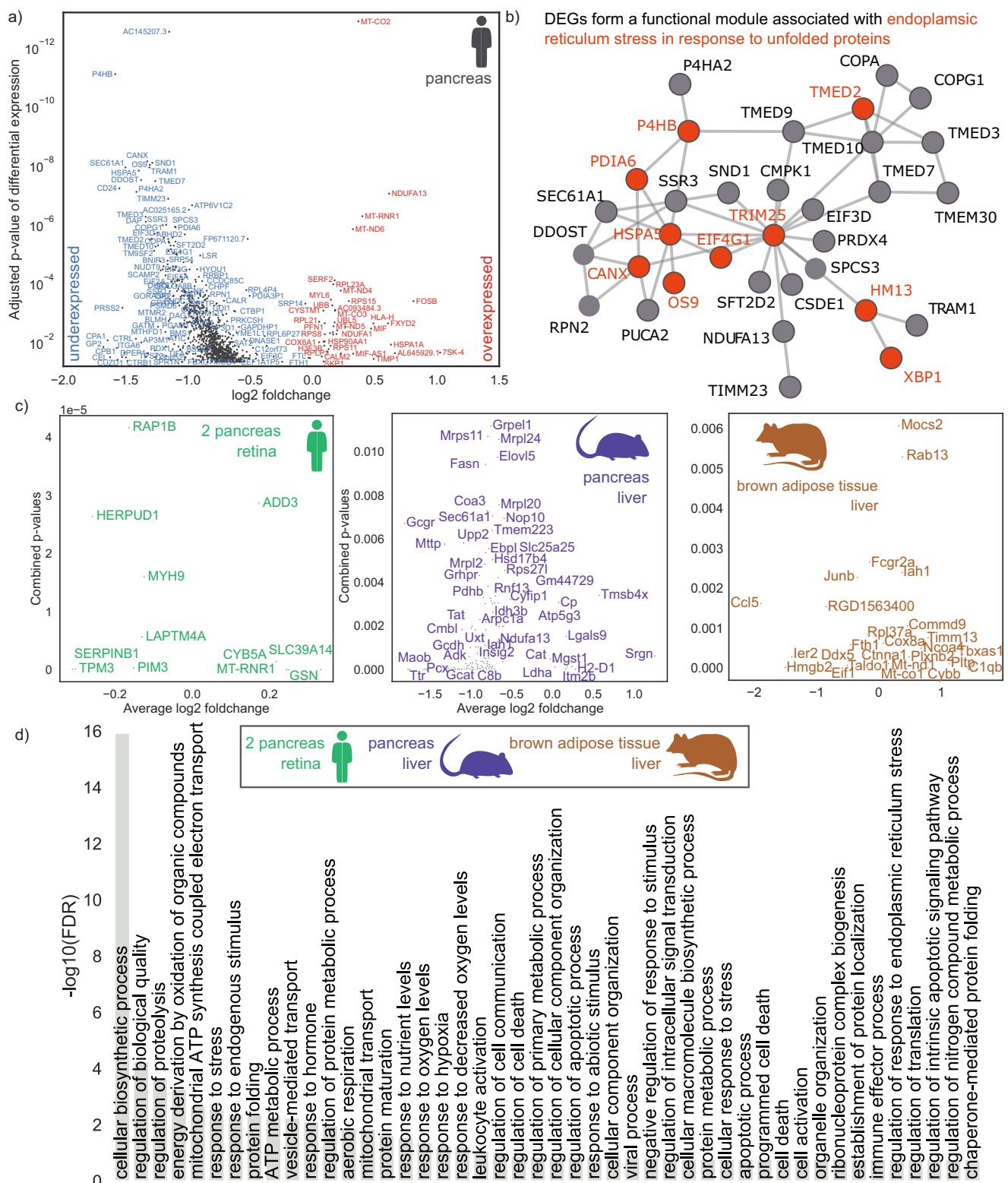

**Fig. 4 | Single-cell transcriptional hallmarks of ageing covary with cryptic mutations. a** Volcano plot of DEGs in cells with cryptic mutations which are not synonymous above 10% heteroplasmy, and those without, in the human pancreas data[30]. Differential expression analysis was done using the Wilcoxon rank-sum test. **b** Genes whose expression covaries with the presence of mutations form a functional protein-interaction module which is enriched with markers of stress response. To obtain the functional module, we combine the *p* values of differential expression with the curated protein interaction network from BIOGRID by computing

a maximum-weight connected subgraph with scPPIN[52]. **c** DEGs in cells with cryptic mutations which are not synonymous for human, mice, and rat. A number of cells in each class is given in Supplementary Table S6. Differential expression analysis was done using the Wilcoxon rank-sum test. **d** Selected GO terms that are enriched in the DEGs of at least six of the seven datasets (two human pancreas, human retina, mouse pancreas, mouse liver, rat liver, rat brown adipose tissue, see Table 1 for details). We order the GO terms by the false discovery rate (FDR) in the data from ref. 30.

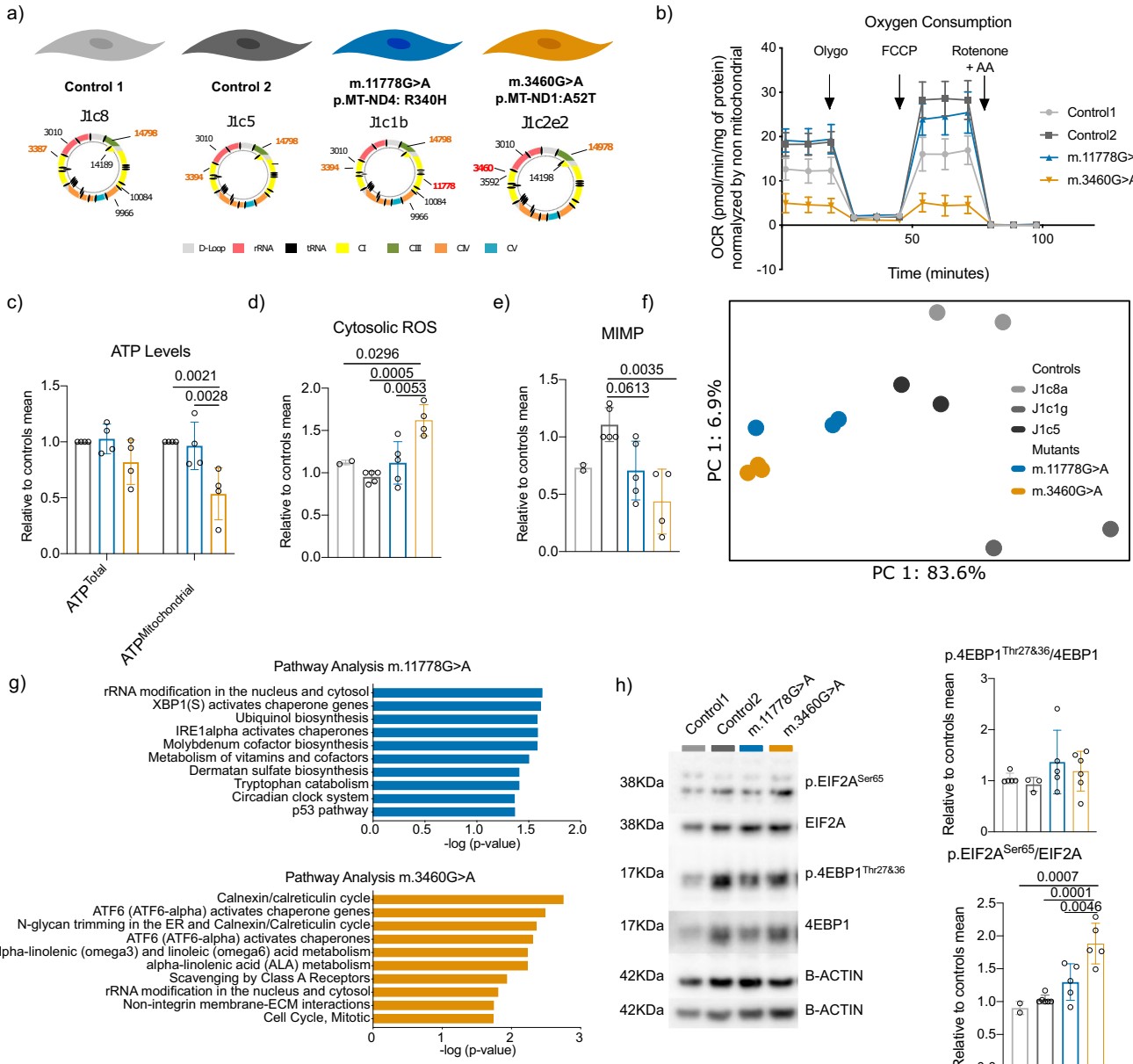

**Fig. 5 | Select cryptic mtDNA mutations can have functional effects including ER-stress. a** Graphical representation of the mtDNA sequences selected for experiment. The mitochondrial population background and location of the mtDNA non-synonymous variants are shown. Full mtDNA sequences are included in Table 3. **b** Oxygen-consumption rate (OCR) from a representative experiment. The Complex V inhibitor oligomycin (Oligo), the uncoupler carbonyl cyanide 4-(tri-fluoromethoxy)phenylhydrazone (FCCP), and the Complex I and III inhibitors antimycin (AA) and rotenone (Rotenone) were added sequentially at the indicated time points. ($n = 3$ for Control 1 and Control 2, $n = 4$ for m.11778G > A and m.3460G > A). **c** ATP-level quantification for total and mitochondrial (glycolysis inhibited by incubation with D-deoxy-glucose instead of glucose). Bars and lines represent mean ± s.d. in all the cell lines. Values are represented relative to the average of control cell lines and all measurements are shown. Statistical testing was performed by using a one-way-ANOVA test followed by Holm–Sidak's multiple comparisons test ($n = 4$). **d** Representative example of FACS determination for cytosolic ROS and quantification of the levels. Bars and lines represent mean ± s.d. in all the cell lines. Values are represented relative to the average of control cell lines and all measurements are shown. Statistical testing was performed by using a one-way-ANOVA test followed by Holm–Sidak's multiple comparisons test ($n = 2$ for Control 1, $n = 5$ for Control 2 and m.11778G > A and $n = 4$ for m.3460G > A). **e** Representative example of FACS determination for MIMP and quantification of

the levels. Bars and lines represent mean ± s.d. in all the cell lines. Values are represented relative to the average of control cell lines and all measurements are shown. Statistical testing was performed by using a one-way-ANOVA test followed by Holm–Sidak's multiple comparisons test ($n = 2$ for Control 1, $n = 5$ for Control 2 and m.11778G > A and $n = 4$ for m.3460G > A). **f** PCA representation of all the DEG genes in both comparisons (DEGs resulting from pair-to-pair comparisons are included in Extended Data). DEGs were found using two-sided Wald test. **g** Mutation-specific pathway analysis (full report is included in Extended Data). **h** Analysis of the EIF2A and 4EBP1 activation. Example of immunoblots. The blots were immunodetected using an anti-EIF2A, anti-p.EIF2ASer51, anti-4EBP1, anti-p.4EBP1Thr37& 46, and β-actin as loading control. Quantification of immune-detected bands for p.EIF2ASer51 and EIF2A as well as anti-p.4EBP1Thr37& 46 and anti-4EBP1corrected with values from the loading control (β-actin) in each cell line. Activation of the kinases is calculated as the ratio of phosphorylated/non-phosphorylated. Bars and lines represent mean ± s.d. in all the cell lines. Values are represented relative to the average of control cell lines and all measurements are shown. Statistical testing was performed by using a one-way-ANOVA test followed by Holm–Sidak's multiple comparisons test. (4EBP1 activation: $n = 5$ for Control 1 and m.11778G > A, $n = 3$ for Control 2, $n = 6$ for m.3460G > A) (EIF2A activation: $n = 2$ for Control 1, $n = 6$ for Control 2, $n = 5$ for m.11778G > A and m.3460G > A).

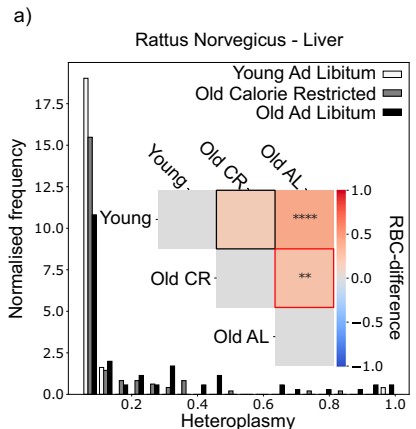
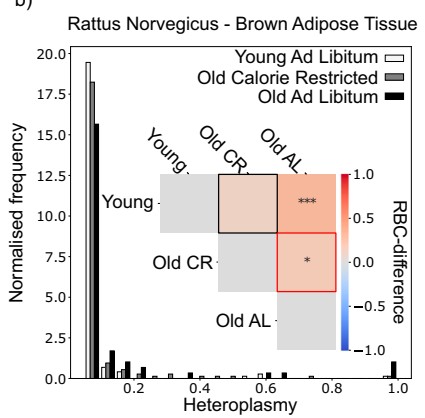
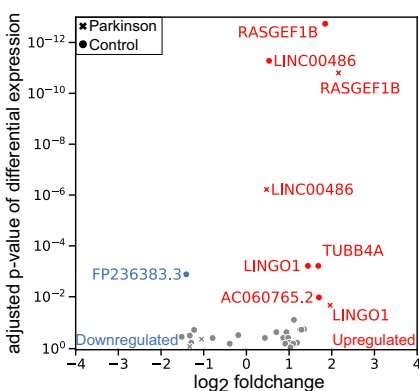

**Fig. 6 | Evidence that cryptic mtDNA mutations have heteroplasmy levels that might be controlled for therapeutic benefit and associations with neurodegeneration-linked genes. a, b** We find the cSFS for rats in three groups, young ad libitum (Y-AL), old ad libitum (O-AL), and old calorically restricted (O-CR), for variants called at a 5% heteroplasmy. This is displayed for liver in (**a**) and for brown adipose tissue in (**b**). Pairwise RBC-difference (see 'Methods' section) between each group's cSFS indicates that caloric restriction slows the accumulation of cryptic mutation. This comparison was done between all pairings of the three groups and is displayed in the inset in (**a**) for liver and for brown adipose tissue in (**b**). Mutations in the O-CR are marginally more likely to be at a higher heteroplasmy than the Y-AL in both the liver and the brown adipose tissue ($p > 0.05$). Compared to this, mutations in the O-AL are far more likely to be at a higher heteroplasmy than Y-AL mice (two-sided Mann–Whitney $U$ test $p < 0.0001$ & $p < 0.001$ Bonferroni corrected in liver and brown adipose tissue, respectively). Critically, mutations in O-AL rats are statistically significantly more likely to be at a higher heteroplasmy than mutations found in O-CR (two-sided Mann–Whitney $U$ test $p < 0.01$ & $p < 0.05$ Bonferroni corrected in liver and brown adipose tissue, respectively). Mean heteroplasmies of observed liver mutations in Y-AL, O-CR, and O-AL, respectively, are $0.0846 \pm 0.0025$, $0.115 \pm 0.0013$, $0.228 \pm 0.0035$. In BAT the means are $0.0796 \pm 0.00068$, $0.0909 \pm 0.00072$, $0.144 \pm 0.0035$. **c** High-heteroplasmy mutational load $\mu^{95\%}$ in single brain cells coincides with differential expression of neurodegeneration-linked genes. Differential expression analysis was done using the Wilcoxon rank-sum test.

($p < 0.05$, Bonferroni corrected). By contrast, the mutations in O-CR rats are not significantly different in heteroplasmy to those found in Y-AL rats. These findings suggest that caloric restriction slows the rate of increase of average cryptic heteroplasmy, this mechanism might be an explanatory factor for the observed longevity in calorically restricted organisms. Caloric restriction can increase the number of mtDNA molecules in rat livers[66]: our theory suggests that increasing mtDNA-copy-number slows the rate of increase of mean heteroplasmy (see Supplementary Discussion S1). An analysis of genes that are differentially expressed in rat cells with cryptic mutations highlighted the differential expression of cystatin C and Apolipoprotein E and enriched GO terms include 'apoptotic mitochondrial changes' and 'ageing'.

**Disease.** Mitochondrial dysfunction is implicated in neurodegenerative diseases such as Parkinson's disease (PD)[67] and Alzheimer's disease (AD)[68]. We use snRNA-seq[69] to investigate whether there is evidence for a perturbation of gene expression in the presence of high-heteroplasmy mtDNA mutations. As snRNA-seq provides sparse coverage of mtDNA transcripts (see Fig. S23) we reduce the minimum coverage for calling heteroplasmies to 10 reads and, to remove potentially falsely called variants, only consider cryptic mutations with a heteroplasmy of at least 95% to identify DEGs in PD and control group, separately (see Fig. 6c). We find three genes significantly upregulated in cells with cryptic mtDNA mutations in both groups: LINGO1 has been associated with various neurodegenerative diseases by inhibiting regeneration in the nervous system[70] and the Guanine nucleotide exchange factor RAP2A has been associated with a population of excitatory neurons in AD[71]. The lncRNA LINC00486 is overexpressed in cells with cryptic mutations and has been associated with common bipolar disorder[72]. For an analysis of AD data that also identifies lncRNAs, specifically MALAT1, see Supplementary Information S5. Further evidence that disease can modulate mitochondrial parameters emerged when we found different mitochondrial ageing rates for diabetic and healthy pancreas tissue (for full results see Supplementary Information S2.7).

## Discussion

We find evidence that an understudied type of single-cell mutation, cryptic mtDNA mutations, while invisible in aggregate, are clock-like predictive of age and markers of ageing and show pathologically-relevant levels of heteroplasmy at middle age and late life. We find evidence that, in post-mitotic tissues, cryptic mutations can evade negative selection, expanding neutrally, and are linked to 5 of 9 hallmarks of ageing[2] (genomic instability, loss of proteostasis, deregulated nutrient-sensing, mitochondrial dysfunction, and altered intercellular communication); specific proteins point to pathways involving mito-protein folding, the ER-stress, and immune responses[73], and we corroborated this with experiment. While the data presented here are from a necessarily limited number tissues, the theory presented in Supplementary Discussion 1 includes discussion of the cell-level parameters most relevant to the observed accumulation, and the impact cell turnover could have on the cSFS. Our formulae predict that the rate of mutation accumulation is increased by increasing mutation rate or mitochondrial turnover and decreased by increasing copy number. The gene-expression changes we observe are consonant with a mosaic combination of the established changes caused by well-studied mitochondrial-disease-associated mtDNA mutations[61]. We conclude with an indication that an anti-ageing therapy can reduce the rate of accumulation of cryptic mtDNA mutations.

## Methods

We construct an analysis pipeline that allows us to identify mtDNA mutations in single cells from single-cell sequencing data. The pipeline enables the analysis of scRNA-seq data (e.g., Smart-seq2, CEL-Seq2, 10x Genomics' Chromium™ Single Cell 3' Solution, and 10x Genomics' Chromium™ Single Cell 3' Solution (single-nuclei protocol), and ATAC-seq) but can be adapted to analyse other sequencing data and is a collection of custom-made SHELL and PYTHON code.

### Data access

**Sequencing data.** We download publicly available sequencing data from the GENE EXPRESSION OMNIBUS with the SRA TOOLKIT, specifically the

**Table 1 | Summary of the public datasets analysed in this study**

| Reference | Species | Tissues | Age range | No. of cells | Accession number | Protocol |
|---|---|---|---|---|---|---|
| Enge et al.[30] | Human | Pancreas | 1 mo–54 yrs | 2544 | GSE81547 | Smart-seq2 |
| Muraro et al.[31] | Human | Pancreas | 23–59 yrs | 12,200 | GSE85241 | CEL-seq2 |
| Voigt et al.[32] | Human | Retinal epithelium and choroid | 54–92 yrs | 8217 | GSE135922 | 10x |
| Almanzar et al.[33] | Mouse | Pancreas, liver | 3–24 mo | 9380 | AWS | Smart-seq2 |
| Corces et al.[34] | Human | Brain tissue | 38–95 yrs | 4835 | GSE147672 | 10x ATAC (single-nucleus) |
| Raredon et al.[35] | Human | Lung | 76–88 yrs | 3630 | GSE133747 | 10x (single-cell) |
| Raredon et al.[35] | Mouse | Lung | 9 wks | 4221 | GSE133747 | 10x (single-cell) |
| Raredon et al.[35] | Rat | Lung | 9 wks | 4429 | GSE133747 | 10x (single-cell) |
| Raredon et al.[35] | Pig | Lung | 3 mo | 1746 | GSE133747 | 10x (single-cell) |
| Buenrostro et al.[43] | Human | Cell lines | N/A | 1344 | GSE65360 | scATAC-seq |
| Ma et al.[65] | Rat | Brown adipose tissue, liver | 5–27 mo | 29,165 | GSE137869 | 10x (single-cell) |
| Smajić et al.[69] | Human | Midbrain | 66–93 yrs | 41,435 | GSE157783 | 10x (single-nucleus) |
| Grubman et al.[91] | Human | Entorhinal cortex | 67–91 yrs | 13,214 | GSE138852 | 10x (single-nucleus) |
| Camunas-Soler et al.[92] | Human | Pancreas | 23–72 yrs | 3789 | GSE124742 | Smart-seq2 |
| | | | | ∑ = 140,435 | | |

Overall, we analyse sequencing data of more than 140,000 cells in four different species and seven different tissues, as well as cell line data.

FASTERQ-DUMP function. The raw read data are in the form of `.fastq` files. For data stored on AWS, we use to AWS CLI to copy `.fastq` files to a local drive. See Table 1 for details on all accession numbers for data in this manuscript.

**Reference genome.** We download reference genome files from https://www.ensembl.org/ or https://www.gencodegenes.org/. Specifically, we download a *gene annotation file* `.gtf` and a *DNA sequence file* `.fa` for each organism. See Table 2 for the genome versions used in this manuscript.

**Alignment, demultiplexing, and UMI counting**
To align raw reads to a reference genome, we use the STAR aligner (version 2.7.5c)[74]. Specifically, for multiplexed data we use STARSOLO, which takes as input the fastq files containing all cells and returns (1) aligned reads in a `.bam` file and also (2) a UMI counts matrix `.tsv`. For non-multiplexed datasets (such as Smart-Seq2) STARsolo appends the sample name to the reads of each cells `.fastq` file and then creates an expression matrix for all cells simultaneously, as well as a `.bam` file which can then be split by sample name for variant calling on each cell.

For full-length datasets, we constructed an expression matrix for the whole experiment with STARsolo, and we created the aligned `.bam` file using STAR for each cell separately. We processed CEL-seq2 `.fastq` files using UMI-Tools (version 1.0.1)[75]. Cellular barcodes of reads were extracted from barcode `.fastq` files and placed into the read names of genomic reads using umi_tools extract. We then aligned reads to the reference genome using STAR with default settings, outputting an aligned `.bam` file. featureCounts was then used to tag reads with the gene they map to. We processed this `.bam` file by using the umi_tools count function to produce an expression matrix. We used the expression matrix and aligned `.bam` files for DEG analysis and variant calling, respectively, in the downstream analysis.

**Variant calling**
We called variants using a custom variant calling pipeline. We attempt to call variants on all cells passing the first round of quality control on the expression matrix. For these cells we import the aligned cellular `.bam` files using pysam (v0.15.3)[76] and search for mutations. To call mutations we first drop reads which are not uniquely aligned to the mitochondrial genome to avoid heteroplasmy calls caused by NUMTs

**Table 2 | Reference genome versions**

| Species | Reference assembly | GTF version |
|---|---|---|
| *Homo sapiens* | GRCh38.p14 | Ensembl release 111 |
| *Mus musculus* | GRCm39 | Ensembl release 110 |
| *Rattus norvegicus* | mRatBN7.2 | Ensembl release 110 |
| *Sus scrofa* | Sscrofa11.1 | Ensembl release 110 |

in the reference genome, then we search the remaining reads to find mismatches between observed bases and the reference base. We only considered genome positions if over 200 reads align to the position with a base quality score above 30. Furthermore, we excluded cells from our analysis if the number of positions passing quality control fell outside of a log-normal distribution, or if the number of positions passing quality control was below 200, so as to exclude cells where we are unlikely to be able to detect any mutations. To calculate heteroplasmy ($h_i$) of a base $i \in [A, C, G, T]$ we take the ratio of the number of reads assigned to that base ($N_i$) to the total number of reads at that position:

$$h_i = \frac{N_i}{\sum_{j \in [A, C, G, T]} N_j}. \tag{1}$$

This definition of heteroplasmy allows for the possibility that two different mutations can occur at the same position on the genome in a cell, which occurs in less than 5% of cells throughout our analysis. To aid comparison between the UMI and non-UMI data, we did not perform deduplication on any dataset. However, we find that when comparisons between heteroplasmy calls are made with and without deduplication on UMI data, strong agreement was observed (see Supplementary Discussion S6.3). Then we classified mutations found in only one cell of a donor/sample at $h_i > 5\%$ as 'cryptic'. We classified any mutations which were common to more than three donors from a given dataset as 'common mutations' and excluded them from further analysis to avoid common RNA variants being used in further analysis. We also exclude any mutation on a site that was not covered with sufficient depth in at least ten cells from a donor (thereby excluding sites with systematically small numbers of reads). After classification, we keep only mutations with $h_i > 10\%$ to exclude possible PCR or sequencing errors (see Supplementary Discussion S6). For rat sequencing data this was done at 5% to enable greater variant

discovery due to the much lower coverage of the mitochondrial genome of that dataset (see Supplementary Fig. S22).

## Mitochondrial information

For human cells, we use the HmtVar[77] database to characterise mtDNA mutations. Specifically, we identify whether mutations result in a non-synonymous amino acid substitution, and classify their pathogenicity by using the MutPred score[78]. MutPred is based on the effect of the substitution on protein structure and function, such that it categorises non-synonomous mutations as either 'low pathogenicity' or 'high pathogenicity'. Combing this information, we obtain three categories of human mutations: 'synonymous', 'low pathogenicity', and 'high pathogenicity'. For mouse cells pathology scoring are unavailable for the majority of mitochondrial encoded proteins, so we classify mutations as either 'synonymous' or 'non-synonymous'.

## Mitochondrial mutation statistics

To quantify the mitochondrial mutation of single cells, we compute different statistics. Given a cell with $m$ mutations at heteroplasmies $\mathbf{h} = (h_1, h_2, ..., h_m)$ with $h_j \in [0, 1]$, we define the *mitochondrial load* as:

$$\mu^{t\%} = \sum_{i=1}^{m} h_j H(h_j - t) \quad (2)$$

where $H(h_j - t)$ indicates the Heaviside step function, such that only heteroplasmies greater than the threshold $t$ contribute to the mutation load. By default, we only count cryptic mutations, which are not synonymous, above 10 % and indicate this for simplicity by $\mu$.

## Quality control

Using the count matrix produced from STARsolo, we exclude cells as recommended by current best practices[79]. For the analysis of the expression matrices, we use SCANPY. We filter each expression matrix using three covariates: the total counts per cell, total genes per cell, and the percentage of reads aligned to the mitochondrial genome. These quality covariates are examined for outlier peaks that are filtered out by thresholding. We determined these thresholds separately for each dataset to account for quality differences. This quality control procedure allows us to establish cells with unexpectedly low read depth or high fraction of mitochondrial content, indicating that mRNA leaked from the cell during membrane permeabilization, or those with high read depth which could be doublets. We only keep cells, which pass both filtering steps, (1) the variant calling filtering (as discussed in subsection 'Variant calling') and (2) the expression matrix filtering (as discussed in this subsection).

## Gene expression analysis

We use SCANPY for single-cell gene expression analysis[80] and perform standard preprocessing steps. In addition to the filtering (as outlined in the 'Quality Control' subsection), we normalised to 10,000 reads per cell and log-transformed the counts. We use a Wilcoxon rank-sum test with a significance threshold of 0.05 for DEG discovery and use the Benjamini–Hochberg procedure to obtain multiple-testing corrected $p$ values. For gene-ontology enrichment and pathway analysis, we use PANTHER[81]. To combine the $p$ value of differential expression with a protein interaction network, we use SCPPIN[52] and visualise it with NETWULF[82]. For bulk sequencing data pathway analyses were performed using the WEB-based GEne SeT AnaLysis Toolkit following their instructions online.

## Cell lines

Mitochondrial cybrids were built using the cell line rho⁰ Cellosaurous 143b.206 (CVL_U923). The original cell line rho⁰ Cellosaurous 143b.206 (CVL_U923) osteosarcoma used to build the hybrids was authenticated[83]. STR profiling were use to authenticate previous cell

## Table 3 | Summary of the cell lines used in this study

| Variant | Haplogroup | GeneBank number |
|---|---|---|
| 11778 | J1c1b | MH080306 |
| 3460 | J1c2e2 | MH080305 |
| Control 1 | J1c8a | JN635299.2 |
| Control 2 | J1c5 | JN635301.2 |
| Control3 | J1c1g | JN635302 |

Haplogroups and accession numbers for bulk cell line data used in Fig. 5a–h.

lines as stated previously[84–86]. Cell lines were grown in DMEM containing glucose (4.5 g l⁻¹), pyruvate (0.11 g/l) and FBS (5%) without antibiotics at 37 °C with 5% $CO_2$. Four cell lines were used two controls and two cell lines carrying mtDNA complex I mutations. All cybrids were obtained from cybrid pools after the selection process[87]. MtDNA sequences of all cell lines can be found in GenBank, and their mtDNA accession numbers are included in Table 3. Replications of these sequences have been carried out in this work through analysis of the full mtDNA sequence from the RNA-seq data.

## Mitochondrial bioenergetics characterisation

Oxygen consumption was performed as previously described[88]. For oxygen-consumption modifications, briefly, $20 \times 104$ cells per well were seeded 8–12 h before measuring basal respiration, leaking respiration, maximal respiratory capacity, and non-mitochondrial respiration (NMR). Respiration levels were determined by adding 1 μM oligomycin (leaking respiration), 0.75 and 1.5 μM carbonyl cyanide-$p$-trifluoromethoxyphenylhydrazone (maximal respiratory capacity) and 1 μM rotenone–antimycin (NMR), respectively. Data were corrected with values from NMR and expressed as pmol of oxygen per min per mg protein.

## Determination of mitochondrial inner membrane potential and cytoplasmic ROS

Determination of mitochondrial inner membrane potential (MIMP) and cytosolic ROS were performed as previously described[88]. Determination of MIMP was carried out using Tetramethylrhodamine, methyl ester at 20 nM in DMSO in parallel with mitochondrial mass detection using MitoTracker Green (20 nM in DMSO). Cytosolic ROS levels were measured using 2′,7′-dichlorofluorescin diacetate at 9 μM in DMSO. All reagents were purchased from Invitrogen. Fluorescence-activated detection was carried using a BD LSRFortessa cell analyzer from BD. A total of 20,000 events were recorded, and doublet discrimination was carried out using FSC-Height and Area in the FlowJo software.

## Determination of ATP levels

ATP levels were determined with previously established methods[87]. ATP levels were measured four times in three independent experiments using the CellTiter-Glo Luminescent Cell Viability Assay (Promega) according to the manufacturer's instructions. Briefly, 10,000 cells per well were seeded, and the medium was changed 48 h before measurement. After that time, cells were lysed, and lysates were incubated with luciferin and luciferase reagents. Samples were measured using a NovoStar MBG Labtech microplate luminometer, and results correspond to the protein quantity measured in a parallel plate.

## Electrophoresis and western blot analysis

Electrophoresis and WB analysis were performed as previously described[88]. Total protein extracts were prepared according to each protein's solubility. Mitochondrial proteins were prepared using 2% dodecyl-maltoside in PBS including protease inhibitors. Protein extracted for kinase phosphorylation analysis was extracted using the PathScan Sandwich ELISA Lysis buffer from Cell Signalling. In any case,

protein extracts were loaded on NuPAGE Bis-Tris Precast mini/midi Protein gels with MES (Invitrogen). Electrophoresis was carried out following the manufacturer's instructions. The SeeBlue Plus2 Pre-stained Protein Standard from Invitrogen was used in each electrophoresis as protein size markers. Separated proteins were transferred to polyvinylidene fluoride membranes. The resulting blots were probed overnight at 4 °C with primary antibodies at the appropriate concentration following the manufacturer's instructions with minor adaptations. The following antibodies were used; anti-EIF2A, Cell Signalling #9722, 1:500; p. anti-p.EIF2ASer51, Cell Signalling #972, 1:500, anti-4EBP1, Abcam ab2606, 1:500; anti-p.4EBP1Thr37&46, Cell Signalling #9459, 1:500; anti-$\beta$-actin, Sigma-Aldrich A5441, 1:2000. After the primary antibody, blots were incubated for 1 h with secondary antibodies conjugated with horseradish peroxidase, and signals were immunodetected using an Amersham Imager 600 and/or medical X-Ray Film blu (Agfa). The bands for each antibody were quantified, aligned and cropped using the Fiji ImageJ 2.3.0/1.53q program, and the O.D. was used as a value for statistical purposes. To avoid interblot variation, one cell line was used as an internal control, and O.D. values corrected for $\beta$-actin levels were shown as relative to the internal control in each case.

### Rank biserial correlation difference

The rank biserial correlation difference, $r$, is a rank correlation. To illustrate its interpretation, consider two sets of heteroplasmic mtDNA mutations, called $G_1$ and $G_2$. State the hypothesis that mutations in $G_1$ are greater in heteroplasmy than mutations in $G_2$. $r$ is an effect size measuring the degree of support for this hypothesis by considering all possible pairings between the mutations in $G_1$ to the mutations in $G_2$. Let $f$ be the proportion of all such pairs of mutations in which mutations in $G_1$ have a greater heteroplasmy than mutations from $G_2$. Let $u$ be the proportion of pairings in which $G_2$ mutations have a greater heteroplasmy than the mutation from $G_1$. The rank biserial correlation is simply defined as $r = f - u$, and can range from $-1$ to $1$. $r = -1$ showing all mutations from $G_2$ have a higher heteroplasmy than mutations from $G_1$, negating the hypothesis. $r = 1$ indicates a positive effect size for the hypothesis indicating that all mutations in $G_1$ are greater in heteroplasmy than those in $G_2$. $r = 0$ indicates the mutations in $G_1$ are observed to have a lower heteroplasmy than $G_2$ mutations as often as they have a higher heteroplasmy[89].

### The Moran model

The theory that we use to forward-model the mtDNA population in a post-mitotic cell is a fixed population size birth–death model with mutation known as the Moran model[36]: it is arguably one of the simplest models that could be chosen. We consider that, at the start of the dynamics, no mutations (unique to each cell) are present in the system: as such, the site frequency spectrum of the system can be seen as out of equilibrium. In order for the population to remain constant at $N$, birth and death events are linked such that every time a randomly chosen mtDNA is replicated, another is randomly chosen to die (the same mtDNA can replicate and then die). As the half-life of mtDNA is much longer than the time taken for replication, we let the time between birth–death events be distributed as $t \sim \text{Exp}(\frac{N(2)}{t_{1/2}})$, where $t_{1/2}$ is the half-life of mtDNA. During every replication, there is a chance that $m$ mutations occur along the length of the replicated mtDNA $L_{\text{mtDNA}}$ due to errors from POLG, and we model this using a binomial distribution of the number of errors $m \sim \text{Binomial}(L_{\text{mtDNA}}, \nu)$ where $\nu$ is the probability of mutation per base of POLG. Through this model, any mutation which enters the system can either be lost, or spread to higher heteroplasmies until it fixes through mtDNA turnover. We make the simplifying assumption that every mutation is unique, and neglect the possibility of back mutation. As the system initially has no mutations present, the mean heteroplasmy of a randomly chosen mutation increases with age. To move beyond a forwards-in-time simulation of the cellular population

of mtDNA we make use of an equivalent backwards-in-time process known as the Kingman coalescent[37] (which captures the behaviour of a wide range of models including the Moran model: our theoretical results are thus not specific to only the Moran model). Full details of this theory, which extends previous work[90] to include inference of the mutation rate, including discussion of the range of forward models the Kingman coalescent applies to, and how the theoretical site frequency spectrum relates to our observed cSFS, are given in Supplementary Discussion 1.

### Reporting summary

Further information on research design is available in the Nature Portfolio Reporting Summary linked to this article.

## Data availability

All analysed data are publicly available from the Gene Expression Omnibus (GEO) website or Amazon Web Services (AWS). The previously published datasets available on GEO are at accession numbers: GSE85241[30], GSE85241[31], GSE135922[32], GSE147672[34], GSE133747[35], GSE65360[43], GSE137869[65], GSE157783[69], GSE138852[91], and GSE124742[92]. The Tabula Muris Senis dataset is available on AWS at https://registry.opendata.aws/tabula-muris-senis/. See Table 1 for a breakdown of cell counts and age ranges for each of the previously published data. The RNA sequencing data generated for this study are available in the GEO database under accession code GSE284767. Processed data are available on GitHub at https://github.com/SystemsAndSignalsGroup/Mito-Ageing[93], as well as code to generate Figs. 1d–g, 2, 3, 4, and 6. Source data for Fig. 5 are provided with this paper. Source data are provided with this paper.

## Code availability

The custom code used for our analysis is available on GitHub at https://github.com/SystemsAndSignalsGroup/Mito-Ageing.

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

## Acknowledgements

We thank the Systems and Signals group at Imperial College London for discussions. This research was funded by Leverhulme (RPG-2018-408), EPSRC (EP/N014529/1), and Wellcome (224486/Z/21/Z). F.K. is supported as an Add-on Fellow for Interdisciplinary Life Sciences by the Joachim Herz Foundation. A.G.D. is a Ramon y Cajal Fellow (RYC2020-029291-I) who receives support from the Spanish Ministry of Science (PID2020-114709RA-I00) and Comunidad de Madrid (Spain) (2019-T1BMD-14236). P.F.C. is a Wellcome Trust Principal Research Fellow (212219/Z/18/Z), and a UK NIHR Senior Investigator, who receives support from the Medical Research Council Mitochondrial Biology Unit (MC_UU_00015/9), the Medical Research Council (MRC) International Centre for Genomic Medicine in Neuromuscular Disease (MR/S005021/1), the Leverhulme Trust (RPG-2018-408), an MRC research grant (MR/S035699/1), an Alzheimer's Society Project Grant (AS-PG-18b-022). This research was supported by the NIHR Cambridge Biomedical Research Centre (BRC-1215-20014). The views expressed are those of the author(s) and not necessarily those of the NIHR or the Department of Health and Social Care.

## Author contributions

N.S.J. conceived and designed the study. A.G., F.K., A.S.M., R.L., J.A., and N.S.J. analysed the data. A.G. and N.S.J. developed the theory. A.G.D. and P.F.C. developed and analysed the cybrid cell lines. All authors interpreted the findings and wrote the manuscript. A.G. and F.K. contributed equally.

## Competing interests

The authors declare no competing interests.
