## [Peer Review File · Nature Communications]

Cryptic mitochondrial DNA mutations coincide with mid-late life and are pathophysiologically informative in single cells across tissues and species

Corresponding Author: Professor Nick Jones

Version 0:

Reviewer comments:

Reviewer #5

(Remarks to the Author)

The authors have suitably addressed most of the reviewer comments. However, several comments remain:

- Figure R4 and the variable and incomplete coverage of these datasets should be commented on in the main text, and figure included in the manuscript or supplement, particularly since most have <20% mtDNA bases covered. These data form the basis of their key findings, and justification for why some samples were excluded from multiple analyses (e.g. the Eye). The authors should also add the tissue that a, b, c correspond to rather than just the sequencing technology (e.g. Pancreas 1 and 2, Eye) to help readers connect this to the other data presented in the main text. They should also use the same y-axis min and max for figures a-d, as is best practice for comparisons, to enable more direct comparison between datasets.

- While the proportion of bases covered was provided in Figure R4, the question on what proportion were shared (vs unique) across the different datasets was not. Can the authors include a plot/data on this, and show what mtDNA bases were sufficiently covered in all human datasets? This is relevant since two of the three main datasets appear to have mostly <20% of the mtDNA covered, and has implications for interpretation of the data particularly if each dataset was capturing distinct areas of the genome.

- I appreciate the authors clarifying that their statement 'after 80 years of age in humans, over 20% of cells carry a mutation with heteroplasmy $h > 95\%$ ' was drawn from Figure 1k, though it remained unclear if their remaining statements were consistent with their data or not – specifically: "estimate of model parameters we see that by 50 years old, over 30% of cells are expected to carry at least one mutation at a heteroplasmy $> 60\%$...30% of cells are expected to carry at least one mutation at homoplasmy is 70 years old" – were these predictions consistent with their single-cell data, i.e. what % of cells at these ages (50 and 70 years) had at least one mutation at heteroplasmy or homoplasmy?

- I recommend the authors add a brief comment in the main text on their rationale for excluding eye samples from subsequent analyses shown in Figure 2 and (most of) Figure 3, to avoid confusion for readers. This is also prudent given nearly all their human samples >60 years of age are from eye.

- "To help guide the reader to this discussion we have now added an additional note in the maintext conclusion" – I could not find the expanded points on effects of cell type/study limitations in the main text conclusion; the authors should cross check and add this in if omitted.

- The authors should use track changes or mark all modifications made in the revised manuscript, as it was difficult to determine what changes were made and where.

(Remarks on code availability)

Reviewer #6

(Remarks to the Author)

The authors have addressed the reviewers thoughtfully. Although the correlation of mtDNA cryptic mutations and aging may not be clear in all tissues, it may be relevant for some tissues.

(Remarks on code availability)

Reviewer #7

(Remarks to the Author)

In this manuscript the authors analyse single-cell transcriptomics data from human, mouse, rat and pig and find that cryptic mtDNA mutations (i.e. mutations that are unique to single cells) accumulate with the age of the donor. Indeed it seems that those mutations make up the majority of mtDNA mutations, but normally escape detection when analysis is performed at the bulk level. The authors also use a random drift model with fixed mtDNA numbers to model the accumulation process of mtDNA mutations over time and also use it to estimate values for the mtDNA mutation rate. The paper has already been extensively reviewed and thus I will keep my comments as short as possible.

I think this is an interesting paper that can help to better understand the role of mtDNA mutations at the single cell level and their role for the aging process. What surprises me most is not so much the extent or accumulation of cryptic mtDNA mutations, but that even those mutations labelled with 'high pathogenicity' seem to expand neutrally, i.e. without counter-selection by the cell. This is an important finding, which in my view needs definitely more room in the Conclusion/Discussion.

If these mutations rise to high heteroplasmy levels and affect the capability of the mitochondrion to produce ATP and/or generate a proton gradient, why are they not detected and removed via the well-known mitochondrial quality control systems (see e.g. Gottlieb et al. 2021)?

And if these point mutations are indeed not removed, why are they not detected in single cell studies like those of e.g. Cao et al. 2001, Gokey et al. 2004, Herbst et al. 2007? Why did these studies find mtDNA deletions but not point mutations?

And finally I wonder why the authors estimated a mutation rate for human mtDNA and not the other species they studied. A comparison would be interesting.

In summary the paper presents an interesting finding regarding the accumulation of cryptic mtDNA point mutations, their apparent accumulation via random drift and I recommend the paper for publication.

(Remarks on code availability)

Version 1:

Reviewer comments:

Reviewer #5

(Remarks to the Author)

The authors have satisfactorily addressed the reviewer comments.

(Remarks on code availability)

Reviewer #7

(Remarks to the Author)

The authors have modified the manuscript to address my concerns and questions regarding their earlier version. As I said, I think the manuscript is an interesting contribution to better understand the role of mtDNA mutations in the aging process and I recommend the manuscript for publication.

(Remarks on code availability)

Response to reviewer comments for “Cryptic mitochondrial ageing coincides with mid-late life and is pathophysiologically informative in single cells across tissues and species”

Referee 5

The authors have suitably addressed most of the reviewer comments. However, several comments remain:

We thank the reviewer for their time reviewing this manuscript, their comments have been helpful in strengthening the clarity of the manuscript and our full responses are given below.

- Figure R4 and the variable and incomplete coverage of these datasets should be commented on in the main text, and figure included in the manuscript or supplement, particularly since most have < 20% mtDNA bases covered. These data form the basis of their key findings, and justification for why some samples were excluded from multiple analyses (e.g. the Eye). The authors should also add the tissue that a, b, c correspond to rather than just the sequencing technology (e.g. Pancreas 1 and 2, Eye) to help readers connect this to the other data presented in the main text. They should also use the same y-axis min and max for figures a-d, as is best practice for comparisons, to enable more direct comparison between datasets.

- While the proportion of bases covered was provided in Figure R4, the question on what proportion were shared (vs unique) across the different datasets was not. Can the authors include a plot/data on this, and show what mtDNA bases were sufficiently covered in all human datasets? This is relevant since two of the three main datasets appear to have mostly < 20% of the mtDNA covered, and has implications for interpretation of the data particularly if each dataset was capturing distinct areas of the genome.

We appreciate that though the the figure comparing coverage was present in the supplement, it was not suitably signposted in the main manuscript. The requested edits to improve the readability of this figure have been made and we have added a sentence highlighting this figure to the main text, and copied it below. We have also updated our supplementary figure S20 showing the coverage across the genome for each dataset to be more clear about the overlap of coverage from each dataset, with the proportion of overlaps being given in this figure caption. We find that due to the 3' nature of the two lower coverage sequencing techniques (CELSEQ2 and 10x) their coverage overlaps consistently with at least 78% of bases in our lowest coverage dataset being covered by the other technologies.

‘Though these sequencing technologies provide differing levels of coverage of the mitochondrial genome (see Supplementary Fig. S20), we find that even when downsampling our high coverage datasets to match the lowest coverage our main conclusions are unchanged (see Supplementary Fig. S24).’

- I appreciate the authors clarifying that their statement ‘after 80 years of age in humans, over 20% of cells carry a mutation with heteroplasmy $h > 95\%$ ’ was drawn from Figure 1k, though it remained unclear if their remaining statements were consistent with their data or not – specifically: “estimate of model parameters we see that by 50 years old, over 30% of cells are expected to carry at least one mutation at a heteroplasmy $> 60\%$...30% of cells are expected to carry at least one mutation at homoplasmy is 70 years old” – were these predictions consistent with their single-cell data, i.e. what % of cells at these ages (50 and 70 years) had at least one mutation at heteroplasmy or homoplasmy?

We have now included a comparison of the modelling predictions to our observed mutation frequencies. We find that the MAP estimate of parameters slightly overestimates the mutation frequency as compared to the data, but the data fall well within the confidence interval of the model.

These predictions are only slightly higher than our observed values, and the data falls well within the confidence intervals of our model. Accounting for coverage differences for donors between 50-60 years old, 22 % of cells are expected to hold a mutation at a heteroplasmy > 60 %, and between 70-80 years old 23 % of cells carry a homoplasmic mutation. At these ages however selective effects may have begun to take effect, removing people with higher than average mutation levels.

- I recommend the authors add a brief comment in the main text on their rationale for excluding eye samples from subsequent analyses shown in Figure 2 and (most of) Figure 3, to avoid confusion for readers. This is also prudent given nearly all their human samples >60 years of age are from eye.

We agree that being clear on our rationale for excluding this data from our analysis in figure 2 is required, and have included a sentence in this section to explain that the limited coverage reduced the power of our tests. For clarity we have linked to supplementary figures S32-S38 where the full analysis of all datasets in figure 3 have been presented.

Due to the lower coverage of the other human datasets (1, 2) we did not have enough mutations to sufficiently power the fisher test for dN/dS ratio or the Mann-Whitney U test for differences in the cSFS.

- “To help guide the reader to this discussion we have now added an additional note in the maintext conclusion” – I could not find the expanded points on effects of cell type/study limitations in the main text conclusion; the authors should cross check and add this in if omitted.

This omission has been corrected in the current version of the manuscript with the following sentence added to the conclusion.

‘While the data presented here are from a necessarily limited number tissues, the theory presented in Supplementary Discussion 1 includes discussion of the cell level parameters most relevant to the observed accumulation, and the impact cell turnover could have on the cSFS.’

- The authors should use track changes or mark all modifications made in the revised manuscript, as it was difficult to determine what changes were made and where.

We apologise for this difficulty, all changes in the manuscript for these responses have been marked in bold.

Referee 6

The authors have addressed the reviewers thoughtfully. Although the correlation of mtDNA cryptic mutations and aging may not be clear in all tissues, it may be relevant for some tissues.

We appreciate the reviewers time addressing the manuscript, and agree that follow on work examining more tissues would be a worthwhile extension.

Referee 7

In this manuscript the authors analyse single-cell transcriptomics data from human, mouse, rat and pig and find that cryptic mtDNA mutations (i.e. mutations that are unique to single cells) accumulate with the age of the donor. Indeed it seems that those mutations make up the majority of mtDNA mutations, but normally escape detection when analysis is performed at the bulk level. The authors also use a random drift model with fixed mtDNA numbers to model the accumulation process of mtDNA mutations over time and also use it to estimate values for the mtDNA mutation rate. The paper has already been extensively reviewed and thus I will keep my comments as short as possible.

We thank the reviewer for their time looking over this paper, and appreciate their comments which we have responded to below.

I think this is an interesting paper that can help to better understand the role of mtDNA mutations at the single cell level and their role for the aging process. What surprises me most is not so much the extent or accumulation of cryptic mtDNA mutations, but that even those mutations labelled with ‘high

pathogenicity' seem to expand neutrally, i.e. without counter-selection by the cell. This is an important finding, which in my view needs definitely more room in the Conclusion/Discussion.

We have expanded the section on the lack of selection for cryptic mutations to more clearly highlight this finding. The discussion now touches on the points below.

If these mutations rise to high heteroplasmy levels and affect the capability of the mitochondrion to produce ATP and/or generate a proton gradient, why are they not detected and removed via the well-known mitochondrial quality control systems (see e.g. Gottlieb et al. 2021)?

Identifying the specific impact of every 'high pathogenicity' mutation is beyond the scope of this paper, and it may be that certain high impact mutations are selectively removed when they occur at very low heteroplasmy levels if they impact a pathway detectable by mitochondrial quality control thresholds. We see evidence for this in the mouse data where the dN/dS ratio suggests negative selection, but there is no difference in the cSFS of synonymous and non-synonymous mutations with heteroplasmy > 10% (see figure 2d). It is unclear however if these mechanisms can detect more mild changes in function caused by point mutations (3).

And if these point mutations are indeed not removed, why are they not detected in single cell studies like those of e.g. Cao et al. 2001, Gokey et al. 2004, Herbst et al. 2007? Why did these studies find mtDNA deletions but not point mutations?

The studies listed are all done in muscle fibres, and focussed on detecting deletions which are the cause of pathology in these fibres. These studies all use PCR based methods to find deletions in the mitochondrial genome by finding mtDNA fragments which are shorter than the expected genome size. Our method used next-generation sequencing technologies, which were not available at the time of these studies, to look at the single-cell level for point mutations accumulating in these cells.

Our full addition to the discussion is shown below:

'The notion of selective mitophagy failing in somatic tissues has been observed in multiple species (3, 4), and while it is known that there are mechanisms by which deletions can expand in muscle fibres (5, 6), these studies have been limited in their ability to detect a similar expansion of point mutations due to their PCR fragment size based approach. Taken together this suggests there is more to be done to understand the role of selective mitophagy in somatic tissues.'

And finally I wonder why the authors estimated a mutation rate for human mtDNA and not the other species they studied. A comparison would be interesting.

We agree that a full fitting to the other species is worthwhile, and have provided full fitting details in Supplementary Discussion S2.5 and Supplementary Figure S11. This discussion has been highlighted in the main text.

In summary the paper presents an interesting finding regarding the accumulation of cryptic mtDNA point mutations, their apparent accumulation via random drift and I recommend the paper for publication.

References

1. M. J. Muraro *et al.*, *Cell Systems* **3**, 385–394.e3, ISSN: 2405-4712 (2016).
2. A. P. Voigt *et al.*, *Proceedings of the National Academy of Sciences of the United States of America* **116**, 24100–24107 (2019).
3. L. C. Greaves *et al.*, *en*, *PLOS Genetics* **8**, ISSN: 1553-7404 (2012).
4. T. E. Kauppila, J. H. Kauppila, N.-G. Larsson, *Cell Metabolism* **25**, 57–71 (2017).
5. A. Herbst *et al.*, *The Journals of Gerontology: Series A* **62**, 235–245, ISSN: 1079-5006 (Mar. 2007).
6. F. Insalata, H. Hoitzing, J. Aryaman, N. S. Jones, *Proceedings of the National Academy of Sciences* **119** (2022).